# OPEN-SET DOMAIN GENERALIZATION FOR SEMANTIC SEGMENTATION

## ABSTRACT

Open-Set domain generalization for semantic segmentation (OSDG-SS) aims to segment known classes and identify unknown categories in target domains that are entirely unseen during training. While recent domain generalization methods perform well under the closed-set assumption, they struggle in open-set settings by misclassifying unknown objects as one of the known classes. To address this challenge, we propose a unified framework that explicitly models unknowns and improves robustness to both semantic and visual domain shifts. First, to provide supervision for unknown regions, we generate realistic unknown objects using Stable Diffusion and insert them into source images, allowing the model to learn unknown-aware representations via segmentation head expansion. However, since synthetic unknowns may not reflect the true distribution of unknowns in target domains, we introduce a meta-learning strategy that partitions the unknown set into meta-train and meta-test subsets, guiding the model to generalize across unseen unknown categories through entropy-based rejection and subdomain shifts. Finally, to reduce confusion between unknowns and visually similar known classes, we optimize the decision boundaries in feature space by enforcing compactness for known classes and expanding the unknown using Mixup-based hard negative synthesis. Extensive experiments across multiple benchmarks demonstrate that our framework significantly improves in the OSDG-SS setting.

## 1 INTRODUCTION

While DG-SS methods have made significant progress, they are typically developed under a closed-set assumption, where the source and target domains are expected to share the same label space. However, in realistic scenarios where target domains are not accessible during training, there is no prior knowledge of which categories may appear at test time. Consequently, target domains may contain novel classes absent from the source domain, a scenario called open-set domain generalization (OSDG). In such cases, models trained under the closed-set assumption are inevitably forced to misclassify unseen classes as one of the known categories, potentially leading to critical failures. As illustrated in Figure 1, we visualize predictions on an unseen target domain that includes unknown classes such as "person" and "traffic sign", which are not present in the source domain. We compare a closed-set DG-SS model (FADA (Bi et al., 2024)) and FADA with an additional post-hoc confidence-based thresholding, which assigns low confidence regions to the unknown class. The FADA baseline (Figure 1(b)) consistently predicts unknown objects as one of the known classes. Even when applying thresholding (Figure 1(c)), it still fails to detect or localize unknown areas, resulting in overconfident and spatially inconsistent predictions. These failure cases reveal the potential risks of deploying closed-set models in the open-set scenario, where unseen classes may appear unexpectedly. This motivates the need for a framework that can explicitly predict unknown classes.

In this work, for the first time, we tackle a novel problem of Open-Set Domain Generalization for Semantic Segmentation (**OSDG-SS**). The goal of OSDG-SS is to train a model on labeled source-domain data such that, when evaluated on unseen target domains, it can (i) *accurately segment pixels belonging to known classes* and (ii) *reliably identify pixels from novel, unseen classes as "unknown"*.

To meet the demands of OSDG-SS, we propose a framework that enables accurate segmentation of known classes and robust detection of unknown classes in unseen target domains. Our approach consists of three key components. First, we compensate for the lack of unknown-class information

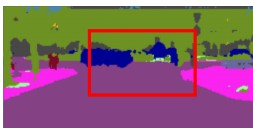 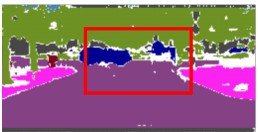 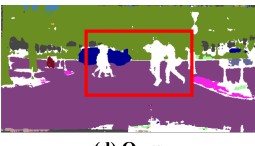

(a) Ground Truth | (b) FADA (NeurIPS' 24) - Closed-Set DG-SS method | (c) FADA with Confidence-based threshold | (d) Ours

Figure 1: Visualization result in the OSDG-SS setting. Given an unseen target domain with unknown classes, existing DG-SS methods such as FADA misclassify unknown regions as known classes, even with confidence-based thresholding, while our method accurately segments unknowns. White represents the unknown classes, and other colors indicate common classes.

in the source domain by synthesizing realistic unknown objects using Stable Diffusion (Rombach et al., 2022), a text-to-image generation model, from textual prompts sampled from an auxiliary object vocabulary (e.g., CIFAR-100 (Krizhevsky et al., 2009) class label set). We then paste the synthesized objects into source domain images for pixel-wise self-supervision. This allows the model to explicitly learn representations of unknown classes by expanding the segmentation head, which includes dedicated unknown-class nodes. Second, we introduce a meta-learning strategy to prevent the model from overfitting to the specific synthetic unknowns used during training. Since these generated samples may differ significantly from the actual unknown classes encountered in target domains, the model must learn to generalize beyond their specific appearances. We split the set of synthetic unknowns into meta-train and meta-test subsets. During meta-train, the model learns to segment synthetic unknowns through explicit supervision. During meta-test, we apply an entropy-based loss that encourages confident activation on unknown-class logits and high uncertainty over known-class logits. This setup enables the model to generalize beyond the training unknowns and recognize previously unseen target classes as "unknown" at the test time. Finally, to reduce confusion between unknowns and visually similar known classes, we apply two complementary regularizations. For known classes, we minimize the Mahalanobis distance between known features and their class-wise prototypes to encourage compactness of known regions. For unknown classes, we synthesize hard negative samples by blending Stable Diffusion-generated images of known and similar unknown classes using Mixup (Zhang et al., 2017). These samples are inserted into training images and learning as unknown. This approach makes the unknown decision boundary broader and more separable, improving robustness to ambiguous unseen unknowns.

Our contributions are summarized as follows:

- We introduce a novel task of Open-Set Domain Generalization for Semantic Segmentation (**OSDG-SS**), where the goal is to segment known classes and simultaneously detect unknowns in unseen domains.
- We introduce an unknown-aware learning framework by synthetic unknown samples using a text-to-image generation model and promote generalizable unknown representations through a meta-learning framework with word-level partitioning, entropy-based rejection, and appearance-level subdomain splits.
- We introduce a feature space regularization scheme that encourages known-class clusters to be compact, and expands unknown regions via Mixup-based synthesis of hard negative samples between known and unknown prototypes.
- Our framework achieves state-of-the-art performance on multiple OSDG-SS benchmarks with large margins, improving both segmentation of known classes and detection of unknowns.

## 2 RELATED WORK

### 2.1 SEMANTIC SEGMENTATION.

Semantic segmentation aims to classify each pixel in an image into a specific semantic category. A foundational approach, Fully Convolutional Networks (FCNs) (Long et al., 2015), has demonstrated impressive performance in this task. To enhance contextual understanding, subsequent works have

introduced methods such as dilated convolutions (Chen et al., 2017), global pooling (Liu et al., 2015), pyramid pooling (Zhao et al., 2017), and attention mechanisms (Zhao et al., 2018; Zhu et al., 2019). More recently, transformer-based methods have achieved significant performance gains (Xie et al., 2021). Despite various studies, semantic segmentation models are still vulnerable to domain shifts or category shifts. To address this issue, we propose a universal domain adaptation for semantic segmentation that handles domain shifts and category shifts.

## 2.2 Domain Generalization for Semantic Segmentation.

Domain Generalization for Semantic Segmentation (DG-SS) aims to train models that generalize well to unseen target domains without any access to target data during training. Early works focused on appearance-level robustness using style transfer (Kim et al., 2023) or frequency adaptation (Bi et al., 2024). Recent approaches leverage powerful vision foundation models (VFMs) such as DINO-V2 (Oquab et al., 2023), CLIP (Radford et al., 2021), SAM (Kirillov et al., 2023), and integrate them with segmentation decoders like Mask2Former (Cheng et al., 2022). Several methods exploit contrastive learning (Choi & Kim, 2024; Wei et al., 2024), hierarchical grouping (Zhang et al., 2023), or self-supervised projection (Li et al., 2023b) to obtain domain-invariant representations. Other lines of work propose transformer-based architectures (Park & Kim, 2024) or network structures such as Siamese learning (Chen et al., 2022) to improve robustness to domain shifts. Some approaches (Ros et al., 2022) leverage diffusion models for domain extension, while others use textual prompts (Kim et al., 2023) to align high-level semantics across domains. Although these approaches improve generalization across domains, they operate under the closed-set assumption and fail to account for the presence of novel categories in the target domain. In contrast, our work addresses open-set domain generalization, where the target may contain previously unseen classes.

## 2.3 Open-Set Domain Generalization.

Open-Set Domain Generalization (OSDG) for classification seeks to learn models that not only generalize across domains but also detect unknown classes in the target domain. Recent methods tackle this by leveraging evidential learning (Zhang et al., 2024), consistency regularization (Zhu & Li, 2021), or data augmentation strategies (Li et al., 2023a). Meta-learning approaches (Peng et al., 2023) have also been proposed to simulate train-test splits between known and unknown categories. These methods often operate at the image or feature level and apply uncertainty-aware objectives to distinguish known from unknown samples. Although progress has been made in classification-level OSDG, these methods are not directly applicable to semantic segmentation, which requires pixel-level predictions with spatial and shape consistency. Our work extends the open-set generalization setting to the semantic segmentation task and introduces an explicit unknown-aware training strategy with realistic object-level unknowns and meta-learning.

## 3 Method

### 3.1 Problem Setup: Open-Set Domain Generalization for Semantic Segmentation

In the OSDG-SS scenario, the goal is to train a segmentation model $f_\theta$ using only labeled source data $\mathcal{D} = \{(x^i, y^i)\}_{i=1}^N$, where $N$ is the total number of samples. Each source image $x^i \in \mathbb{R}^{H \times W \times 3}$ is an RGB image, and $y^i \in \mathbb{R}^{H \times W \times C}$ is the corresponding pixel-wise ground-truth label. At test time, the model is evaluated on unseen target domains and is expected to segment pixels from known classes and classify pixels from novel classes, which are not present in the source label space, as "unknown".

To tackle this scenario, we propose three key components: (i) generating and learning from realistic unknown samples, (ii) a meta-learning strategy for unknown generalization, and (iii) optimizing decision boundaries for robust unknown detection. Figure 2 illustrates the overall architecture.

### 3.2 Meta-learning for Unknown Generalization (MUG)

The challenge of open-set domain generalization lies in enabling the model to identify unknown classes it has never seen during training. Rather than relying solely on thresholding or energy-based post-hoc rejection, we propose a learning strategy that leverages synthetic unknown samples

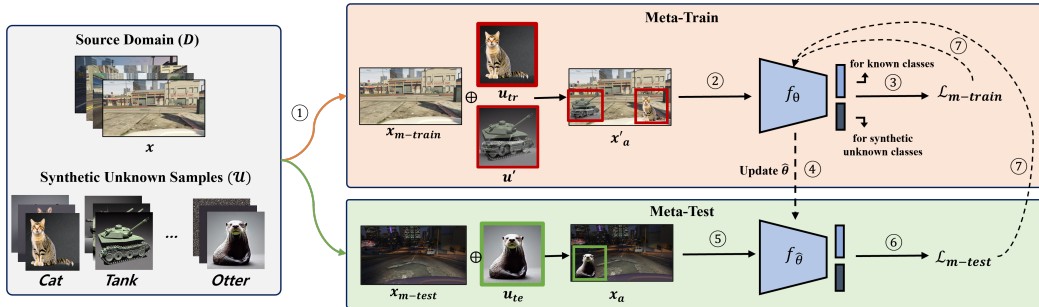

Figure 2: **Overview of our proposed framework.** We split the source domain $\mathcal{D}$ and the synthetic unknown samples set $\mathcal{U}$ into disjoint meta-train and meta-test subsets. During meta-train (top), we generate hard negatives by mixing synthetic unknowns $\mathcal{U}_{\text{tr}}$ with their most similar known objects, then paste them into source images to form augmented samples $x'_a$ for $\mathcal{L}_{m-train}$. During meta-test (bottom), we use unseen source images and unknown samples $\mathcal{U}_{\text{te}}$ to compute an $\mathcal{L}_{m-test}$. The numbers indicate the sequence of the training process.

while applying meta-learning to simulate an open-set scenario. Specifically, we expose the model to semantically meaningful unknowns that resemble objects outside the source label space, but at the same time encourage generalization beyond these specific categories to avoid overfitting. This strategy enables the model to learn a generalized concept of "unknown" and effectively handle unseen unknowns at deployment.

We first construct an unknown word set $\mathcal{U} = \{u_1, \ldots, u_K\}$ by sampling class names from an external vocabulary (e.g., CIFAR-100), excluding overlap with known and target classes. These prompts are fed into Stable Diffusion (Rombach et al., 2022) to generate realistic object-shaped unknowns. Each generated image is processed with a pretrained segmentation model (e.g., SAM Kirillov et al. (2023)) to extract object masks. The segmented unknown objects are then composited onto source-domain images, forming augmented training samples. Corresponding pixel-level labels are updated to assign the pasted region to a newly defined unknown class.

To reflect the discrepancy between training-time and real-world unknowns, we partition $\mathcal{U}$ into disjoint subsets: $\mathcal{U}_{\text{tr}} = \{u_1^{\text{tr}}, \ldots, u_L^{\text{tr}}\}$ for meta-train and $\mathcal{U}_{\text{te}} = \{u_1^{\text{te}}, \ldots, u_{K-L}^{\text{te}}\}$ for meta-test. During meta-train, we generate unknowns using $\mathcal{U}_{\text{tr}}$ and expand the segmentation head to $C + L$ classes. The model is supervised using cross-entropy loss:

$$\mathcal{L}_{\text{seg}} = -\sum_{j=1}^{H \cdot W} \sum_{c=1}^{C+L} y(j,c) \log f_\theta(x)(j,c), \tag{1}$$

where $C$ is the number of known classes and $L = |\mathcal{U}_{\text{tr}}|$. During meta-test, we present samples containing unseen unknowns from $\mathcal{U}_{\text{te}}$, and apply an entropy-based loss to guide the model toward rejecting these as "unknown". Specifically, for pixels $x_i \in X_{\text{unk}}$, the loss encourages low confidence on known logits and high confidence on unknown logits:

$$\mathcal{L}_{\text{ent}} = -\frac{1}{|X_{\text{unk}}|} \sum_{x_i \in X_{\text{unk}}} \left[ \sum_{c=1}^{C} p_c(x_i) \log p_c(x_i) - \sum_{c=C+1}^{C+L} p_c(x_i) \log p_c(x_i) \right]. \tag{2}$$

Here, $p_c(x_i)$ is the softmax probability of pixel $x_i$ belonging to class $c$. This formulation allows the model to learn the concept of "unknown" as a general semantic category, decoupled from specific identities. To further simulate appearance-level domain shift, we split source images into two sub-domains based on average RGB brightness. Meta-train and meta-test samples are drawn from different sub-domains, exposing the model to both semantic and low-level visual variation.

More importantly, the meta-learning procedure enhances the generalization ability to previously unseen unknowns. By explicitly separating synthetic unknowns into Meta-Train and Meta-Test subsets, the model is repeatedly exposed to a setting where the training and testing of unknown categories do not overlap. In this way, the model cannot rely on memorizing the appearance of specific

synthetic samples, but instead must capture The broader and transferable concept of "unknown" that distinguishes it from known classes. During meta-train, the model acquires pixel-level supervision on synthetic unknowns, while during meta-test, it learns to reject novel, unseen unknowns through entropy-based regularization. This combination allows the model to form a generalized representation of unknowns, leading to improved prediction performance on entirely unseen categories in the target domain, even when their visual or semantic properties differ significantly from the synthetic unknowns used in training.

### 3.3 Optimizing Decision Boundaries for Robust Unknown Detection

Although the above training strategy equips the model with unknown awareness, misclassification can still occur when unseen unknown classes resemble known ones in appearance. For instance, a novel object with a similar texture or shape may be incorrectly predicted as a known class. To mitigate this, we propose a decision boundary optimization framework to reshape the feature space.

We aim to (i) compress known-class clusters for clear class separation, and (ii) expand the unknown region to occupy broader, non-overlapping areas in the embedding space. Two complementary strategies are employed.

**Prototype-Based Compactness for Known Classes.** We define a compactness loss over all pixels of known classes using Mahalanobis distance from the class prototype $\mu_c$:

$$\mathcal{L}_{\text{proto}} = \frac{1}{C} \sum_{c=1}^{C} \frac{1}{|\mathcal{X}_c|} \sum_{x_i \in \mathcal{X}_c} \sqrt{(f(x_i) - \mu_c)^T \Sigma_c^{-1} (f(x_i) - \mu_c)}, \tag{3}$$

where $\mathcal{X}_c$ is the set of features from class $c$, and $\Sigma_c$ is the covariance matrix. This encourages tighter known-class clusters.

**MixUp-Based Expansion for Unknowns.** To increase separation between unknowns and visually similar knowns, we identify for each known class $c$ its most similar unknown prototype $\mu_u$ and generate a hard negative by mixing two Stable Diffusion outputs:

$$u^* = \arg\max_{u \in \mathcal{U}_{\text{tr}}} \text{sim}(\mu_c, \mu_u), \quad u' = \text{MixUp}(\text{SD}(c_{\text{word}}), \text{SD}(u_{\text{word}}^*)), \tag{4}$$

and assign the resulting region the label of $u^*$ only. The pasted sample $(x'_a, y'_a)$ is supervised via:

$$\mathcal{L}_{\text{mix}} = - \sum_{j=1}^{H \cdot W} \sum_{c=1}^{C+L} y'_a(j, c) \log f_\theta(x'_a)(j, c). \tag{5}$$

Together, these losses promote compact known decision boundaries and broad unknown coverage, reducing ambiguity near class borders and enhancing open-set robustness.

### 3.4 Optimizing Decision Boundaries for Robust Unknown Detection (ODB)

While the components described above enable the model to detect unknown classes, confusion can still occur when an unseen unknown class closely resembles a known class in appearance. For example, an unknown object with a texture or structure similar to a known class may be mistakenly classified as that class. To mitigate this issue, we propose a decision boundary optimization strategy that reshapes the feature space so that known classes form compact clusters, while unknown classes are encouraged to form broad and non-overlapping regions that do not interfere with known-class boundaries. To achieve this, we apply two complementary regularization techniques during training: (i) prototype-based compactness for known classes, and (ii) hard negative sample generation for expanding the unknown region.

**Prototype-Based Compactness for Known Classes (PCK).** We newly define a prototype compactness loss over all pixels belonging to each known class, using Mahalanobis distance to measure deviation from the class prototype. For the class $c$, we compute:

$$\mathcal{L}_{\text{proto}} = \frac{1}{C} \sum_{c=1}^{C} \frac{1}{|\mathcal{X}_c|} \sum_{x_i \in \mathcal{X}_c} d_{\text{M}}(x_i), \tag{6}$$

---

**Algorithm 1** Meta-Learning Procedure for OSDG-SS

---

**Require:** Source domain $\mathcal{D}$, known classes $\mathcal{C}$, unknown words $\mathcal{U}$, model parameters $\theta$;
         learning rates $\alpha, \beta, \eta$
**Ensure:** Updated parameters $\theta$
1: Split $\mathcal{U} \rightarrow \mathcal{U}_{\mathrm{tr}}, \mathcal{U}_{\mathrm{te}}$
2: Split $\mathcal{D} = \{(x, y)\}$ into $\mathcal{D}_{\mathrm{br}}^{\mathrm{high}}, \mathcal{D}_{\mathrm{br}}^{\mathrm{low}}$ by mean brightness of images
3: **while** $\theta$ not converged **do**
4:     **Meta-Train**:
5:     Generate unknowns from $\mathcal{U}_{\mathrm{tr}}$ and Mixup-based hard negatives, and paste them into $\mathcal{D}_{\mathrm{br}}^{\mathrm{high}}$
6:     Compute $\mathcal{L}_{m-train} \leftarrow \mathcal{L}_{\mathrm{mix}} + \mathcal{L}_{\mathrm{proto}}$
7:     Update: $\hat{\theta} \leftarrow \theta - \alpha\nabla_\theta\mathcal{L}_1$
8:     **Meta-Test**:
9:     Generate unseen unknowns from $\mathcal{U}_{\mathrm{te}}$ and paste them into $\mathcal{D}_{\mathrm{br}}^{\mathrm{low}}$
10:    Compute $\mathcal{L}_{m-test} \leftarrow \mathcal{L}_{\mathrm{seg}} + \mathcal{L}_{\mathrm{proto}} + \mathcal{L}_{\mathrm{ent}}$
11:    Update: $\theta \leftarrow \theta - \eta(\nabla_\theta\mathcal{L}_1 + \beta\nabla_{\hat{\theta}}\mathcal{L}_2)$
12: **end while**

---

where $d_{\mathrm{M}}(x_i) = \sqrt{(f(x_i) - \mu_c)^T \Sigma_c^{-1}(f(x_i) - \mu_c)}$ is the Mahalanobis distance between the pixel feature $f(x_i)$ and the prototype $\mu_c$. This encourages all known-class features to lie close to their respective prototypes, promoting global intra-class compactness and sharper decision boundaries.

**Hard Negative Mixup for Unknown Region Expansion (MEU).** We aim to expand the unknown decision boundary while maintaining a clear separation from known classes, in order to prevent visually similar unknown objects from being misclassified as known classes. To this end, we introduce a Mixup-based strategy that synthesizes ambiguous unknown samples designed to challenge the model. For each known class $c$, we compute its prototype $\mu_c$ and identify the most similar unknown prototype $\mu_u$ based on cosine similarity:

$$u^* = \arg \max_{u \in \mathcal{U}_{\mathrm{tr}}} \mathrm{sim}(\mu_c \cdot \mu_u). \tag{7}$$

We generate synthetic images for the known class $c$ and its most similar unknown class $u^*$ using Stable Diffusion, guided by the corresponding text prompts $c_{\mathrm{word}}$ and $u^*_{\mathrm{word}}$. These images are blended via Mixup to create a hard negative sample:

$$u' = Mixup(SD(c_{\mathrm{word}}), SD(u^*_{\mathrm{word}})), \tag{8}$$

where $SD(\cdot)$ denotes the output of Stable Diffusion from a class-specific prompt. Importantly, the resulting sample is assigned the label of the unknown class $u^*$ only, without mixing class labels. This sample is then pasted into the source image to form a new augmented sample $(x'_a, y'_a)$, which is trained using the standard cross-entropy loss:

$$\mathcal{L}_{\mathrm{mix}} = -\sum_{j=1}^{H \cdot W} \sum_{c=1}^{C+L} y'_a(j, c) \log f_\theta(x'_a)(j, c). \tag{9}$$

This strategy encourages the unknown nodes to occupy a broader region of the feature space, reducing overlap with known classes and improving robustness to unseen but similar unknowns.

To sum up, these two regularizations reshape the feature distribution such that known classes are tightly enclosed, and unknown classes, especially those visually similar to known ones, are assigned to their own distinct region. This structure reduces open-set confusion and enables the model to reject challenging unknowns more reliably during inference.

### 3.5 OVERALL TRAINING PROCEDURE

We summarize the entire learning procedure in Algorithm 1, which iteratively trains the model with synthetic unknowns, meta-learning over unseen unknowns, and decision boundary optimization. During meta-train, the model is updated using the Mixup loss $\mathcal{L}_{\mathrm{mix}}$ and the prototype loss $\mathcal{L}_{\mathrm{proto}}$. During meta-test, the model is evaluated using the segmentation loss $\mathcal{L}_{\mathrm{seg}}$, the prototype loss $\mathcal{L}_{\mathrm{proto}}$, and the entropy loss $\mathcal{L}_{\mathrm{ent}}$ to simulate generalization to unseen unknowns under appearance shift.

Table 1: Semantic segmentation performance on the OSDG-SS benchmarks. We report results trained on GTA5 and SYNTHIA, and evaluated on Cityscapes, BDD100k, and Mapillary, under the open-set domain generalization setting. Our method outperforms existing approaches across both common and private classes. The best results are highlighted in bold.

| Method | Backbone | Trained on GTA5 | | | | | | | | | Average | | |
| | | → Cityscapes | | | → BDD100k | | | → Mapillary | | | | | |
| | | Common | Private | H-Score | Common | Private | H-Score | Common | Private | H-Score | Common | Private | H-Score |
|---|---|---|---|---|---|---|---|---|---|---|---|---|---|
| *Open-Set Domain Generalization for Classification based:* | | | | | | | | | | | | | |
| CrossMatch Zhu & Li (2021) (CVPR' 21) | ResNet-50 | 31.30 | 14.21 | 19.54 | 23.31 | 6.89 | 10.63 | 28.54 | 4.50 | 7.77 | 27.71 | 6.32 | 10.20 |
| MEDIC Wang et al. (2023) (ICCV' 24) | ResNet-50 | 41.34 | 8.82 | 14.53 | 32.71 | 8.97 | 14.08 | 37.23 | 4.25 | 7.63 | 37.09 | 7.34 | 12.08 |
| EBiL-HaDS Peng et al. (2024) (NeurIPS' 24) | ResNet-50 | 39.79 | 7.06 | 11.99 | 15.25 | 5.60 | 8.19 | 13.84 | 3.77 | 5.92 | 22.96 | 5.47 | 8.70 |
| *Closed-Set Domain Generalization for Semantic Segmentation based:* | | | | | | | | | | | | | |
| IBN-Net Pan et al. (2018) (CVPR' 22) | ResNet-50 | 22.64 | 18.76 | 20.51 | 19.34 | 9.94 | 13.13 | 16.92 | 7.83 | 10.70 | 19.63 | 12.17 | 14.78 |
| TLDR Kim et al. (2023) (ICCV' 23) | ResNet-50 | 52.58 | 11.32 | 18.63 | 44.89 | 8.57 | 14.39 | 30.73 | 5.42 | 9.21 | 42.73 | 8.43 | 14.08 |
| BlindNet Ahn et al. (2024) (CVPR' 24) | ResNet-50 | 34.77 | 9.00 | 14.30 | 30.39 | 7.45 | 11.96 | 33.21 | 8.90 | 14.04 | 32.79 | 8.45 | 13.43 |
| Rein Wei et al. (2024) (CVPR' 24) | DINO-V2 | 54.24 | 12.37 | 20.14 | 49.58 | 7.51 | 13.04 | 46.21 | 9.51 | 15.77 | 50.01 | 9.79 | 16.32 |
| TQDM Pak et al. (2024) (ECCV' 24) | DINO-V2 | 54.39 | 15.05 | 23.84 | 48.51 | 9.81 | 23.84 | 50.84 | 14.84 | 22.97 | 52.24 | 13.23 | 21.04 |
| FADA Bi et al. (2024) (NeurIPS' 24) | DINO-V2 | 57.43 | 17.83 | 27.21 | 51.80 | 8.74 | 14.96 | 51.29 | 16.04 | 24.43 | 53.51 | 14.20 | 22.20 |
| *Open-Set Domain Generalization for Semantic Segmentation (Ours) based:* | | | | | | | | | | | | | |
| **Ours** | DINO-V2 | **61.84** | **30.28** | **40.65** | **54.48** | **18.37** | **27.48** | **53.64** | **24.26** | **33.41** | **56.65** | **24.30** | **33.84** |

| Method | Backbone | Trained on SYNTHIA | | | | | | | | | Average | | |
| | | → Cityscapes | | | → BDD100k | | | → Mapillary | | | | | |
| | | Common | Private | H-Score | Common | Private | H-Score | Common | Private | H-Score | Common | Private | H-Score |
|---|---|---|---|---|---|---|---|---|---|---|---|---|---|
| *Open-Set Domain Generalization for Classification based:* | | | | | | | | | | | | | |
| CrossMatch Zhu & Li (2021) (CVPR' 21) | ResNet-50 | 20.51 | 37.23 | 4.25 | 13.13 | 22.60 | 6.55 | 12.40 | 14.40 | 5.85 | 15.35 | 24.74 | 5.55 |
| MEDIC Wang et al. (2023) (ICCV' 24) | ResNet-50 | 25.59 | 37.58 | 7.06 | 16.35 | 36.32 | 6.24 | 10.65 | 22.94 | 6.92 | 17.53 | 32.28 | 6.74 |
| EBiL-HaDS Peng et al. (2024) (NeurIPS' 24) | ResNet-50 | 36.27 | 7.68 | 34.14 | 38.56 | 6.76 | 6.34 | 34.34 | 6.27 | 10.60 | 36.39 | 6.90 | 17.03 |
| *Closed-Set Domain Generalization for Semantic Segmentation based:* | | | | | | | | | | | | | |
| IBN-Net Pan et al. (2018) (CVPR' 22) | ResNet-50 | 32.50 | 7.20 | 12.00 | 28.40 | 5.80 | 9.50 | 25.90 | 5.10 | 8.60 | 28.93 | 6.03 | 10.03 |
| TLDR Kim et al. (2023) (ICCV' 23) | ResNet-50 | 35.00 | 8.10 | 13.00 | 30.20 | 6.20 | 10.00 | 28.40 | 5.90 | 9.30 | 31.20 | 6.73 | 10.77 |
| BlindNet Ahn et al. (2024) (CVPR' 24) | ResNet-50 | 39.32 | 10.24 | 16.25 | 38.52 | 7.24 | 12.18 | 34.34 | 7.10 | 10.67 | 37.39 | 8.19 | 14.97 |
| Rein Wei et al. (2024) (CVPR' 24) | DINO-V2 | 44.51 | 12.71 | 19.77 | 40.33 | 11.77 | 18.12 | 41.59 | 13.48 | 20.35 | 42.14 | 12.65 | 16.41 |
| TQDM Pak et al. (2024) (ECCV' 24) | DINO-V2 | 47.68 | 12.68 | 20.03 | 41.83 | 11.53 | 18.70 | 43.26 | 12.38 | 20.63 | 44.26 | 12.20 | 17.79 |
| FADA Bi et al. (2024) (NeurIPS' 24) | DINO-V2 | 49.34 | 14.42 | 22.14 | 44.08 | 12.55 | 19.62 | 46.52 | 16.00 | 23.67 | 46.64 | 14.32 | 19.81 |
| *Open-Set Domain Generalization for Semantic Segmentation (Ours) based:* | | | | | | | | | | | | | |
| **Ours** | DINO-V2 | **55.70** | **20.90** | **30.07** | **48.47** | **14.77** | **24.69** | **50.44** | **19.88** | **28.52** | **51.53** | **19.01** | **27.76** |

# 4 EXPERIMENTS

## 4.1 EXPERIMENTAL SETUP

**Datasets.** We evaluated our method on two newly defined OSDG-SS benchmarks using synthetic-to-real settings. Specifically, we used GTA5 (Richter et al., 2016) and SYNTHIA (Ros et al., 2016) as labeled source domains, and Cityscapes (Cordts et al., 2016), BDD100K (Yu et al., 2020), and Mapillary (Neuhold et al., 2017) as target domains.

**Evaluation Protocols.** In the OSDG-SS setting, it is essential to evaluate both segmentation performance on common classes and unknown classes. We use H-Score as the evaluation metric, defined as the harmonic mean of the Common mIoU (mean Intersection-over-Union) and the Private ("unknown") IoU. This provides a balanced measure of performance in open-set scenarios.

**Implementation Details.** Our method is built upon FADA (Bi et al., 2024), a frequency-adapted domain generalization method. We adopt the same training backbone and decoder setup as FADA. We adopt DINO-V2 (Oquab et al., 2023) as the Vision Foundation Model (VFM) backbone and use a Mask2Former (Cheng et al., 2022) decoder for segmentation. For unknown sample generation, we construct an unknown word set by excluding both the known classes and the target unknown classes from CIFAR-100, resulting in 88 remaining class names. These are used as prompts for Stable Diffusion to synthesize realistic unknown objects, ensuring that no overlap occurs with the target unknown categories. During meta-learning, we split both the word-based unknown class set and the RGB-based image classification cues into meta-train and meta-test subsets with a ratio of 3:1. The learning rates for the meta-learning stage are set to $\eta = 0.001$, $\alpha = 0.01$, and $\beta = 0.01$. Regarding the computational time of using the diffusion model, we have used a single RTX 3090 GPU, and it takes about 3 seconds to generate one image with the diffusion model. However, since we generate a synthetic image from text prompts in advance before training begins, there is no additional computational cost from using the diffusion model during the training process.

**Baselines.** Since no prior work has addressed the OSDG-SS setting, we adapt existing methods from related domains for comparison. First, for OSDG for image classification methods (Zhu & Li, 2021; Peng et al., 2024; Wang et al., 2023), we experimented by changing the backbone to a

Table 2: Ablation study of our method trained on GTA5. Baseline is FADA with confidence thresholding. The best results are highlighted in bold.

| Config. | Method | Trained on GTA5 | | | | | | | | | Average | | |
| | | → Cityscapes | | | → BDD100k | | | → Mapilliary | | | | | |
| | | Common | Private | H-Score | Common | Private | H-Score | Common | Private | H-Score | Common | Private | H-Score |
|---|---|---|---|---|---|---|---|---|---|---|---|---|---|
| A | FADA with Confidence-based threshold | 57.43 | 18.59 | 27.21 | 51.80 | 8.74 | 14.97 | 51.29 | 17.81 | 24.43 | 53.50 | 15.71 | 24.08 |
| B | A + MUG (w/o meta-learning) | 52.91 | 20.49 | 29.54 | 49.34 | 14.02 | 21.84 | 50.34 | 18.56 | 27.12 | 50.86 | 17.69 | 26.16 |
| C | B + MUG (full) | 59.42 | 26.43 | 36.60 | 52.29 | 16.78 | 25.41 | 51.98 | 21.20 | 30.12 | 54.56 | 21.47 | 30.71 |
| **Ours** | C + ODB | **61.84** | **30.28** | **40.65** | **54.48** | **18.37** | **27.48** | **53.64** | **24.26** | **29.10** | **56.65** | **24.30** | **33.84** |

semantic segmentation model. In this case, we used the DeepLabv2 (Chen et al., 2017) segmentation network and ResNet-50 (He et al., 2016) as the backbone. For the CSDG-SS methods (Pan et al., 2018; Ahn et al., 2024; Kim et al., 2023; Wei et al., 2024; Pak et al., 2024; Bi et al., 2024), we apply confidence-based thresholding (Choe et al., 2024) to detect unknown regions, making them compatible with the OSDG-SS setting.

## 4.2 COMPARISONS WITH THE BASELINES

We compared our method against a wide range of adapted baselines, including open-set domain generalization (OSDG) methods originally designed for classification, and closed-set domain generalization methods for semantic segmentation. Table 1 shows results under two settings: training on GTA5 (Top) and SYNTHIA (Bottom), and testing on three real-world target domains—Cityscapes, BDD100K, and Mapillary. Open-set classification methods such as CrossMatch (Zhu & Li, 2021), MEDIC (Peng et al., 2024), and EBiL-HaDS (Wang et al., 2023) are adapted to the semantic segmentation task. These methods show limited ability to handle dense spatial predictions and often underperform in both common and private regions. Closed-set DG-SS methods including IBN-Net (Pan et al., 2018), BlindNet (Ahn et al., 2024), and FADA (Bi et al., 2024) are extended to the open-set setting using post-hoc confidence-based thresholding (Choe et al., 2024). While they achieve competitive results on common classes, their ability to handle unknown classes remains limited, as reflected by lower H-Scores. In contrast, our method achieves the best performance across all target domains and training settings, consistently outperforming all baselines in both common and private class accuracy. Notably, our method achieves an average H-Score of 33.84 when trained on GTA5 and 27.76 when trained on SYNTHIA, demonstrating superior robustness to domain and category shifts. These results validate the effectiveness of our framework in a realistic OSDG-SS scenario.

## 4.3 ABLATION STUDY

**Ablation Study about Proposed Framework.** We conducted an ablation study to evaluate the contribution of each component in our framework. Table 2 shows results using the GTA5 → Cityscapes, BDD100K, Mapillary setting. Config. A represents the baseline, where FADA (Bi et al., 2024) is extended with confidence-based thresholding to reject low-confidence regions. In Config .B, we utilize a synthetic image train as unknown class, which explicitly trains the model to recognize realistic unknown samples generated via Stable Diffusion. This improves the average H-Score from 22.20 to 26.16, demonstrating the benefit of learning from object-shaped unknowns. Config. C further incorporates our meta-learning strategy (MUG), which simulates novel unknowns during training and pushes the model to generalize beyond seen unknown categories, resulting in an H-Score of 30.70. Finally, our full model adds the decision boundary optimization module (ODB), leading to the best performance of 33.85. These results highlight the complementary effect of unknown-aware learning, meta-learning, and feature space regularization in addressing the OSDG-SS challenge.

**Ablation Study about MUG.** We conducted an ablation study to evaluate the effectiveness of each component in our meta-learning strategy. As shown in Table 3, the baseline corresponded to applying unknown sample learning on top of FADA, achieving an average H-Score of 26.16. Introducing meta-learning with only the unknown word set split improved performance to 28.03. Further applying domain split based on brightness led to an additional gain (29.05), demonstrating the benefit of simulating domain-level variation. Finally, incorporating the entropy-based loss $\mathcal{L}_{ent}$ into the meta-test phase resulted in the best performance across all target domains, reaching an

Table 3: Ablation study on MUG. The best results are highlighted in bold.

| MUG | | | Trained on GTA5 | | | Average |
|---|---|---|---|---|---|---|
| | | | $\rightarrow$ Cityscapes | $\rightarrow$ BDD100k | $\rightarrow$ Mapillary | |
| $\mathcal{U}$ split | $\mathcal{D}$ split | $\mathcal{L}_{ent}$ | | H-Score | | |
| | | | 29.54 | 21.84 | 27.12 | 26.16 |
| ✓ | | | 32.15 | 23.41 | 28.54 | 28.03 |
| ✓ | ✓ | | 34.28 | 24.45 | 28.42 | 29.05 |
| ✓ | ✓ | ✓ | **36.60** | **25.41** | **30.12** | **30.71** |

Table 4: Ablation study on ODB. The best results are highlighted in bold.

| ODB | | Trained on GTA5 | | | Average |
|---|---|---|---|---|---|
| | | $\rightarrow$ Cityscapes | $\rightarrow$ BDD100k | $\rightarrow$ Mapilliary | |
| PCK | MEU | | H-Score | | |
| | | 36.60 | 25.41 | 30.12 | 30.71 |
| ✓ | | 38.95 | 23.13 | 32.84 | 31.73 |
| | ✓ | 37.45 | 25.95 | 31.64 | 31.68 |
| ✓ | ✓ | **40.65** | **27.48** | **33.41** | **33.84** |

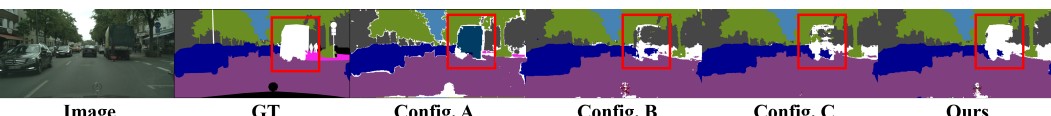

|  Image | GT | Config. A | Config. B | Config. C | Ours |

Figure 3: Qualitative results of our method trained on GTA5 and evaluated on Cityscapes. White represents the unknown classes, while other colors indicate common classes.

average H-Score of 30.71. These results confirmed that each component contributed to improved generalization to unseen unknowns in the OSDG-SS setting.

**Ablation Study about ODB.** Table 4 shows the impact of the individual components of ODB, namely PCK and MEU trained on GTA5. The baseline included unknown training and meta-learning, yielding an average H-Score of 30.71. Applying prototype-based compactness (PCK) improved the alignment of known-class features by reducing intra-class variance near the decision boundary, resulting in 31.73. Applying Mixup-based expansion for unknowns (MEU) encouraged the model to distinguish unknowns from visually similar known classes by expanding the unknown feature region, leading to 31.68. When both were applied together, the model benefited from tighter known boundaries and more separable unknown distributions, achieving the best performance of 33.84. These results confirmed that both the local compactness of known features and the strategic expansion of unknowns played complementary roles in improving robustness to unknown regions in open-set segmentation.

### 4.4 QUALITATIVE RESULT

To better understand the contribution of each component in our framework, we provide qualitative comparisons across the ablation configurations shown in Table 2. As illustrated in Figure 3, Config A (FADA with confidence thresholding) often fails to localize unknown regions, either missing them entirely or producing fragmented masks. Adding synthetic unknowns (Config. B) improves coverage but still suffers from misclassification. Config. C, which includes our meta-learning strategy, captures more of the unknown region but remains noisy near boundaries. Ours, which incorporates decision boundary optimization, produces the most accurate and spatially coherent predictions, demonstrating the synergistic effect of our proposed components in OSDG-SS.

## 5 CONCLUSION

We presented a unified framework for Open-Set Domain Generalization for Semantic Segmentation (OSDG-SS), addressing the realistic and challenging scenario where target domains contain unknown classes not observed during training. To this end, we introduce a new framework (i) explicit unknown supervision via Stable Diffusion-based synthesis, (ii) a meta-learning strategy that promotes generalizable unknown representations by simulating unknown-class shifts through partitioned supervision and entropy-based loss, and (iii) feature space regularization to compact the decision boundary for known classes and expand the decision boundary for unknown classes. Extensive experiments on synthetic-to-real benchmarks demonstrated that our approach consistently outperforms prior DG-SS and OSDG classification methods adapted to segmentation, both in segmenting known classes and detecting unknowns. We hope this work provides a foundation for future research on unknown-aware generalization in semantic segmentation.

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
