# OpenReview forum: "Open-Set Domain Generalization for Semantic Segmentation"
_ICLR.cc/2026/Conference — ICLR 2026 Conference Withdrawn Submission_

### Official Review · Reviewer_5QDD · 2025-10-30

**Soundness:** 2
**Presentation:** 2
**Contribution:** 2
**Rating:** 2
**Confidence:** 4

**Summary:**

The paper proposes a framework for Open-Set Domain Generalization for Semantic Segmentation. It utilizes Stable Diffusion to generate objects of the unknown class and introduces a meta-learning strategy to guide the model to generalize across unseen unknown categories through entropy-based rejection and subdomain shifts. Extensive experiments on synthetic-to-real benchmarks demonstrated the effectiveness of the proposed approach.

**Strengths:**

1. The proposed method shows the effectiveness of the proposed framework on open-set domain generalization segmentation task.
2. Ablation study shows the effectiveness of the component of the proposed framework.
3. The readability of this paper is good, and the structure is clear.

**Weaknesses:**

1. The performance improvement may rely on the number of unknown class names used.
2. Some open-vocabulary segmentation methods also investigate the performance in the cross-domain setting. Missing comparison with the segmentation methods designed for the open-set setting, such as open-vocabulary segmentation methods.
3. The content of section 3.3 is highly similar to the content of 3.4. The formula (3) is the same as (6); The formula (4) is the same as the combination of (7) and (8); The formula (5) is the same as (9).
4. The techniques, e.g., mixup, prototype, and meta learning, are common in DG. The paper seems to be a combination of common techniques.  Thus, I question the contribution/novelty to the level of ICLR.

**Questions:**

1.  How the common/private class splits are defined and how many different splits are used in the experiment?
2. If there is a overlap between the set of used unknown class names with the set of private class names?

---

### Official Review · Reviewer_HipR · 2025-11-01

**Soundness:** 2
**Presentation:** 2
**Contribution:** 3
**Rating:** 4
**Confidence:** 4

**Summary:**

This paper formalizes OSDG-SS, training only on labeled source data to segment known classes and detect unseen ones as “unknown.” It introduces a unified approach that synthesizes unknown objects and pastes them into source images, uses meta-learning with semantic splits and an entropy objective to generalize beyond specific synthetic unknowns, and shapes features to tighten known clusters while expanding unknown regions. On several datasets, the method consistently surpasses adapted open-set classification and closed-set DG-SS baselines.

**Strengths:**

- The paper presents a novel and impactful formulation of OSDG-SS with training-time unknown modeling instead of post-hoc rejection.
- The meta-learning design and feature-space shaping are well motivated.
- The approach achieves strong and consistent gains on benchmarks using appropriate open-set metrics and provides clear ablations that substantiate each component’s contribution.

**Weaknesses:**

- The analysis of the synthetic-unknown design is limited, as the paper does not thoroughly study sensitivity to vocabulary and prompting - choices, the proportion and difficulty of unknowns.
- Despite leading quantitative metrics on both common and unknown classes, the visualization results suggest noticeably degraded segmentation quality for some known categories; the paper does not analyze what causes these errors—e.g., boundary shifts from hard-negative Mixup, over-regularization from prototype compactness, or confusion induced by entropy training—nor does it provide additional quantitative and qualitative comparisons to clarify this discrepancy.

**Questions:**

Please refer to my weakness section.

---

### Official Review · Reviewer_7F5Z · 2025-11-01

**Soundness:** 1
**Presentation:** 2
**Contribution:** 2
**Rating:** 2
**Confidence:** 5

**Summary:**

This paper defines the novel problem of Open-Set Domain Generalization for Semantic Segmentation. This task requires a model trained only on labeled source domains to generalize to unseen target domains, where it must both accurately segment known classes and reliably identify all novel, unseen classes as "unknown". The authors demonstrate that existing Domain Generalization (DG) methods, which operate under a closed-set assumption, fail in this setting by misclassifying unknown objects as known classes.

**Strengths:**

1. To compensate for the lack of unknown-class data, the framework generates realistic unknown objects using Stable Diffusion from an auxiliary vocabulary (e.g., CIFAR-100) and pastes them into source images, providing explicit pixel-level supervision for a dedicated unknown class.
2. To prevent the model from overfitting to the specific synthetic unknowns, a meta-learning strategy is introduced. The set of synthetic unknowns is partitioned into meta-train and meta-test subsets. The model learns from the meta-train set with standard supervision and is guided to reject the unseen meta-test set using an entropy-based loss, promoting a more general representation of "unknown". This is combined with a simple appearance-based subdomain split to simulate domain shifts.
3. A feature space regularization scheme is proposed to reduce confusion between known classes and visually similar unknowns. This includes (a) Prototype-Based Compactness to tighten known-class feature clusters using Mahalanobis distance and (b) Mixup-Based Expansion to create hard-negative samples by mixing generated images of known classes and their most similar unknown prototypes, thereby expanding the unknown decision boundary.
4. Experiments on new OSDG-SS benchmarks show that the proposed method significantly outperforms adapted OSDG classification baselines and closed-set DG-SS methods

**Weaknesses:**

1. Section 3.3 ("OPTIMIZING DECISION BOUNDARIES...") and Section 3.4 ("OPTIMIZING DECISION BOUNDARIES...")  are identical.
2. The use of CIFAR-100  as the source for unknown prompts seems poorly matched to the target domains (urban driving scenes). The unknowns in Figure 1 are "person" and "traffic sign", which are not in CIFAR-100. While MUG is meant to generalize, starting with a more relevant vocabulary (e.g., non-target classes from COCO) seems more logical.
3. The paper frames OSDG-SS by combining three distinct challenges: segmentation, domain generalization, and open-set recognition. However, one might question the practical significance of this specific problem formulation in the current landscape. With the rapid advancement of powerful, large-scale foundation models for segmentation, which are trained on web-scale data, the combined challenge of domain shift and open-set rejection may be substantially mitigated or rendered less critical. Furthermore, the proposed method is validated on relatively small-scale synthetic-to-real datasets (e.g., GTA/SYNTHIA to Cityscapes). This validation is insufficient to conclusively demonstrate the method's effectiveness, especially when the core premise is to build a model robust enough for real-world deployment where the variety of domains and unknown objects is far greater.

**Questions:**

The most important question is weakness3. Is this problem still necessary?

---

### Official Review · Reviewer_Q2v7 · 2025-11-07

**Soundness:** 2
**Presentation:** 3
**Contribution:** 2
**Rating:** 2
**Confidence:** 3

**Summary:**

The paper proposes a framework to address the open-set domain generation for semantic segmentation. This makes use of a meta-learning procedure to train a model for better modelling of the unknown category in the embedding space, where a prototype-based loss and synthetic unknown are generated using diffusion models are also leveraged. The paper claims to be the first work on open-set domain generation for semantic segmentation. However, it is not clear how it is different from pixel-based semantic segmentation with OOD objects. The proposed method is only compared with the domain generalization methods but not compared with the methods proposed for semantic segmentation with OOD objects.

**Strengths:**

It is important to develop semantic segmentation methods which can detect unknown classes. The proposed framework is a reasonably motivated and designed.

**Weaknesses:**

1. The way they frame the problem is similar to pixel-based semantic segmentation with OOD objects. There have been a number of recent works published in recent years. It is important to state clearly how it is different from those related work. Just a few examples of the related work:

- W. Zhao, J. Li, X. Dong, Y. Xiang and Y. Guo, "Segment Every Out-of-Distribution Object," in Proceedings of 2024 IEEE/CVF Conference on Computer Vision and Pattern Recognition (CVPR), Seattle, WA, USA, 2024, pp. 3910-3920, doi: 10.1109/CVPR52733.2024.00375.
- H. Choi, H. Jeong and J. Y. Choi, "Balanced Energy Regularization Loss for Out-of-distribution Detection," in Proceedings of 2023 IEEE/CVF Conference on Computer Vision and Pattern Recognition (CVPR), Vancouver, BC, Canada, 2023, pp. 15691-15700, doi: 10.1109/CVPR52729.2023.01506.
- Huachao Zhu, Zelong Liu, Zhichao Sun, Yuda Zou, Gui-Song Xia, Yongchao Xu; “Beyond Pixel Uncertainty: Bounding the OoD Objects in Road Scenes” in Proceedings of the IEEE/CVF International Conference on Computer Vision (ICCV), 2025, pp. 8472-8481

2. For the methods being proposed, it is not sure how its performance is related to the number of unknown classes selected for the modelling the unknown categories (Eq. 1&2), even though a loss function is introduced for regularization (Eq. 2).

3. The Mix-Up (Eq. 4) seems to be a crucial component and the author may want to provide a bit more details. With the use of stable diffusion, I think the computational cost will be increased accordingly. Related information and discussion should be included.

4. Section 3.4 is an expanded version of Section 3.3, with almost the same title and contents.

5. It is important to compare with the existing semantic segmentation methods with OOD objects.

**Questions:**

Q1: How is the problem you are working on different from the existing work on pixel-based semantic segmentation with OOD objects?

Q2: How sensitive is the proposed method against the number of classes selected for modeling the unknown category?

Q3: Does the version of stable disfusion used for computing the Mix-up matter for the overall performance?

---

### Official Review · Reviewer_Cncj · 2025-11-08

**Soundness:** 2
**Presentation:** 2
**Contribution:** 2
**Rating:** 4
**Confidence:** 3

**Summary:**

This paper proposes Open-Set Domain Generalization for Semantic Segmentation (OSDG-SS), a framework designed to segment known classes while rejecting unseen ones without using any target domain data. The authors synthesize object-like unknowns through Stable Diffusion and Segment Anything Model (SAM) to mimic real-world unseen categories. A meta-learning strategy is then introduced, dividing these unknowns based on semantic and visual attributes (e.g., word and brightness) to emulate various unseen cases. To further refine generalization, the model incorporates prototype compactness and MixUp based hard negative sampling. Built on DINO-V2 and Mask2Former, OSDG-SS consistently achieves higher H-Scores than prior domain generalization and open-set segmentation baselines across multiple benchmarks, including SYNTHIA.

**Strengths:**

1. The study clearly defines a novel setting that integrates open-set recognition within domain generalization and adopts the H-Score for evaluation.
2. The training flow is coherently structured, progressing from text-to-image synthesis via Stable Diffusion, followed by SAM-based masking, compositing, meta-splitting, and entropy-based rejection.
3. Decision boundary refinement is mathematically grounded through prototype cohesion, MixUp hard negatives, and Mahalanobis-distance based regularization.
4. Synthetic data generation is handled offline, effectively avoiding additional training-time overhead.

**Weaknesses:**

1. The method depends too much on SD-generated objects that do not reflect real unknowns like textures or lighting changes.
2. Different backbones and only simple thresholding are used for baselines, so the real improvement of the method is uncertain.
3. All experiments are synthetic-to-real and only H-Score is reported. Broader real-to-real validation and threshold-free metrics are needed.
4. Pre-generation of synthetic data is costly, and several typos or notation mismatches reduce clarity (e.g., Mapilliary -> Mapillary)

**Questions:**

N.A

---

### Note · Authors · 2025-11-20

I have read and agree with the venue's withdrawal policy on behalf of myself and my co-authors.